# A Near-Real-Time Flood Detection Method Based on Deep Learning and SAR Images

Xuan Wu [1,2,3], Zhijie Zhang [4,5,*], Shengqing Xiong [5], Wanchang Zhang [1,2], Jiakui Tang [6,7], Zhenghao Li [1,2,3], Bangsheng An [1,2,3] and Rui Li [1,2,3]

1   Key Laboratory of Digital Earth Science, Aerospace Information Research Institute, Chinese Academy of Sciences, Beijing 100094, China
2   International Research Center of Big Data for Sustainable Development Goals, Beijing 100094, China
3   University of Chinese Academy of Sciences, Beijing 100049, China
4   School of Geography, Development and Environment, The University of Arizona, Tucson, AZ 85719, USA
5   Natural Resources Aerogeophysical and Remote Sensing Center of China Geological Survey, Beijing 100083, China
6   College of Resources and Environment, University of Chinese Academy of Sciences, Beijing 100049, China
7   Yanshan Earth Key Zone and Surface Flux Observation and Research Station, University of Chinese Academy of Sciences, Beijing 101408, China
*   Correspondence: zhangzhijie@uarizona.edu

**Abstract:** Owning to the nature of flood events, near-real-time flood detection and mapping is essential for disaster prevention, relief, and mitigation. In recent years, the rapid advancement of deep learning has brought endless possibilities to the field of flood detection. However, deep learning relies heavily on training samples and the availability of high-quality flood datasets is rather limited. The present study collected 16 flood events in the Yangtze River Basin and divided them into three categories for different purpose: training, testing, and application. An efficient methodology of dataset-generation for training, testing, and application was proposed. Eight flood events were used to generate strong label datasets with 5296 tiles as flood training samples along with two testing datasets. The performances of several classic convolutional neural network models were evaluated with those obtained datasets, and the results suggested that the efficiencies and accuracies of convolutional neural network models were obviously higher than that of the threshold method. The effects of VH polarization, VV polarization, and the involvement of auxiliary DEM on flood detection were investigated, which indicated that VH polarization was more conducive to flood detection, while the involvement of DEM has a limited effect on flood detection in the Yangtze River Basin. Convolutional neural network trained by strong datasets were used in near-real-time flood detection and mapping for the remaining eight flood events, and weak label datasets were generated to expand the flood training samples to evaluate the possible effects on deep learning models in terms of flood detection and mapping. The experiments obtained conclusions consistent with those previously made on experiments with strong datasets.

**Keywords:** near-real-time flood detection; synthetic aperture radar; deep learning; convolutional neural network; Yangtze River basin



## 1. Introduction

Flooding is one of the most devastating natural hazards, causing economic losses of about USD 25.5 billion and 6570 fatalities worldwide annually on average between 1970 and 2020 [1]. The property and life losses related to flooding have accelerated at a rate of 6.3% and 1.5% per year, respectively, over the past five decades [2], and the global economic losses caused by flooding are projected to increase by 17% over the next 20 years [3]. China is a seriously affected country frequently faced with flood disasters, with huge economic losses and high fatalities [4,5]. For example, the flooding that took

place in 2020 over southern China affected 30.2 million people, with an economic loss of about CNY 61.79 billion. Near-real-time flood mapping becomes a very necessary action to cope with flood rescue and disaster assessment with advancement of earth observation technologies by satellites.

Satellite-based flood mapping provides an effective means for near-real-time flood detection, which can accurately describe the dynamic processes of flooding in both the temporal and spatial scales [6]. Compared with ground observations, satellite-based observations have unique advantages in flood detection and mapping, as they are quick, accurate, and cover an extensive area. Flood detection using optical remote sensing is mainly based on spectral information to detect waterbodies caused by undulation through normalized difference water index (NDWI) [7–9] or other segmentation algorithms [10–12]. Although some satisfactory achievements have been made with these methods, inherent limitations are yet involved in optical remote sensing-based flood detection due to its daytime-only operating mode and weak cloud penetrating capability. Synthetic aperture radar (SAR) can work under all-day and all-weather conditions, providing data support for near-real-time flood detection [13,14]. The global threshold method is an efficient and convenient solution for flood mapping using SAR images [15,16]. However, due to the complex characteristics of SAR images, the accurate detection of floods by image segmentations with a single threshold is very difficult [17]. Threshold algorithms based on regional difference have been proposed [11,18,19]. In addition, some automatic threshold algorithms, such as Otsu [20,21], entropy threshold [22], and bimodal histogram [23] algorithms, are widely used for flood detection. The undulated area by flooding can be detected very effectively with the change detection method from flood and non-flood images. As an efficient and convenient image segmentation approach, threshold method is very suitable for large-scale, near-real-time flood detection. However, the threshold method cannot deal with complex nonlinear problems, and has a lack of spatial consistency and is vulnerable to noise interference [24–26]. Therefore, many studies combine the change detection approach with the threshold method to obtain different image-specific information, and then use the threshold method to extract the changed part [27–29]. Nevertheless, both the threshold method and change detection approach rely heavily on expert knowledge and require tedious satellite image preprocessing [30]. Additionally, most flood detection methods are aimed at a single flood event; they cannot be transferred and reused on other flood events.

Traditional flood detection methods are labor-intensive and time-consuming, dependent on expert knowledge and face a lack of portability and scalability in most cases. In recent decades, deep learning, especially convolutional neural networks (CNNs), has made great achievements in remote sensing applications [31]. Convolutional neural network is an end-to-end efficient self-learning model, and has been widely used in automatic flood detection [32]. A flood detection dataset released for deep learning based on images from Sentinel 1 and 2 by Bonafilia et al. [33] was evaluated with various CNNs focusing on performances of those CNNs [26,34,35]. Performances of various CNNs compared with those obtained by traditional threshold methods in flood detection of Poyang Lake by Dong et al. suggested that CNNs can effectively suppress the speckle noise of SAR images [36]. Subsequently, an effective self-learning CNN model was proposed and applied to the urban area of Houston, USA for flood detection by Li et al. [37]. Although great achievements have been made in flood detection by using remotely sensed data with CNNs, several challenges are yet to be solved in near-real-time flood detection, especially at large-scale detections, i.e.,:

(1) As a data-driven algorithm, deep learning for flood detection lacks the support of big data;
(2) Generation of training data for deep learning is currently a labor-intensive and time-consuming task. Discovering a method to efficiently generate representative training datasets for deep learning is an issue worth studying;
(3) Most flood detection methods developed in the past are aimed at a single flood event, but they are difficult to transfer and reuse for other flood events.

The performance of the satellite-based flood detection and mapping for an individual flood events may be affected by sensors, satellite attitudes, or atmospheric conditions, etc., and flood training samples obtained from multiple flood events can be used with a deep learning model to minimize errors introduced by those effects. To address these issues, the present study took the Yangtze River Basin (YRB) as an experimental study region to investigate the possibility of the application of SAR Images with a deep learning model for developing a near-real-time flood detection and automatic mapping approach. The main contents and highlights of present study can be summarized as follows:

(1) An efficient and fast approach for generating a standard flood training dataset for flood detection with deep learning was proposed;
(2) Two kinds of standard flood training datasets generated by the proposed approach, namely a strong and weak labeled dataset, were used to evaluate the performances of several CNNs;
(3) Large-scale flood detection in the YRB was attempted with deep learning models.

The paper is structured as follows: Section 2 introduces the study area, satellite data, and dataset production along with the method for dataset generation. Section 3 presents the models proposed or adopted and the performance of each model trained with the strong label dataset as well as the flood detection results of the Yangtze River basin. The performances of the models trained with the weak label dataset, the perspective on the change detection methods and some limitations of present study are discussed in Section 4. The conclusions made from this study are given at the end of the paper.

## 2. Materials and Methods

### 2.1. Study Area and Data

Rising in the Tanggula Mountains in west-central China, the Yangtze River is about 3964 miles (6380 km) long and flows from its source in a glacier in Qinghai Province, eastwards into the East China Sea at Shanghai, receiving water from over 700 tributaries along the way with catchment area of about 1.8 million $km^2$ in China. Under the influence of monsoon climate, the YRB has long been subject to an uneven temporal–spatial distribution of precipitation and temperature with a great inter-annual variation and concentrated intra-annual distribution, one of the most important factors with respect to frequent flooding. Floods occur almost every year in the Yangtze River basin. Many obvious anomalous changes in the spatial–temporal distribution have been observed in recent decades compared with the past, which may very likely upset the established balance between the existing river runoff and flood control system and result in unexpected major disasters. In the present study, for developing a near-real-time flood detection and automatic mapping approach by using remote sensing with the deep learning model, 16 flood events that took place in the past decade in the YRB were systematically investigated, and a total of 32 Sentinel-1 SAR images acquired in flood and non-flood periods were used as main satellite data sources in this study. Since only backscattering intensity data is needed for SAR image processing, Ground Range Detected (GRD) data of Sentinel 1 was used in this study. GRD data includes VH and VV polarization data, which are represented later in VH and VV.The 12.5 m DEM generated from ALOS-PALSAR was used as auxiliary data for CNN model training. To meet the requirements of near-real-time flood detection, 16 images acquired in 8 flood events were used for the training and testing of the deep learning models, and the remaining 16 images derived from the remaining 8 flood events were used for flood detection. Table 1 lists the locations and flooding durations of the 16 selected flood events, as well as the IDs of Sentinel-1 SAR images acquired in corresponding flood events along with their usages in present study.

**Table 1.** Information about the selected flood events and the corresponding Sentinel-1 SAR images used in present study.

| Flood Events | Flood Period | Image ID | | Train or Test |
| --- | --- | --- | --- | --- |
| Dongting Lake | 9 June 2016–3 July 2016 | 011CB0_5A05 | 0127C5_F17D | Train and Test |
| Poyang Lake | 30 May 2016–17 July 2016 | 011822_A928 | 012E7C_86E8 | Train and Test |
| Middle Reaches of the Yangtze River | 11 June 2016–5 July 2016 | 011D9A_0801 | 0128B9_886D | Train and Test |
| Poyang Lake | 12 June 2017–6 July 2017 | 00A8F1_7632 | 00B2FB_3091 | Train and Test |
| Juzhang River | 5 July 2018–29 July 2018 | 02747C_139D | 027F52_9A16 | Train and Test |
| Huaihe River | 7 August 2018–19 August 2018 | 02836E_FDCB | 028919_2CEB | Train and Test |
| Middle Reaches of the Yangtze River | 2 July 2019–14 July 2019 | 032778_40DE | 032CC4_6DEF | Train and Test |
| Ruan Jiang | 30 July 2020–11 August 2020 | 03E75D_6DAE | 03ED1D_5ADE | Test |
| Dongting Lake | 4 June 2017–10 July 2017 | 01C150_503B | 01D14B_23A9 | Application |
| Poyang Lake | 20 June 2020–26 July 2020 | 029F8B_298B | 02AF8A_BF3A | Application |
| Chaohu Lake | 3 July 2020–27 July 2020 | 03DB5D_91FD | 03E612_6A3E | Application |
| Fujiang River | 14 August 2020–19 September 2020 | 03EE9F_8BAF | 04012C_41B2 | Application |
| Dongting Lake | 19 June 2020–25 July 2020 | 03D52B_49F4 | 03E52E_6E8E | Application |
| Middle and Lower Reaches of the Yangtze River | 14 June 2020–8 July 2020 | 03D2E0_90D3 | 03DD85_0B97 | Application |
| Middle and Lower Reaches of the Yangtze River | 14 June 2020–8 July 2020 | 03D2E0_261F | 03DD85_725A | Application |
| Upper Reaches of the Yangtze River | 16 August 2021–21 September 2021 | 04A272_97F5 | 04B46E_4D61 | Application |

An overview of the YRB with the geo-location of the Sentinel-1 SAR images used in the present study is shown in Figure 1.

*2.2. Method*

The flowchart of present study, as presented in Figure 2, consists of three parts. The first part mainly involves the satellite data preprocessing. Six steps of preprocessing, i.e., orbit correction, thermal noise removal, radiometric calibration, speckle filtering, terrain corrections, and decimalization, were applied to the Sentinel-1 images acquired in each of the 8 selected flood events in the YRB, Meanwhile, DEM corresponding to the coverage of each Sentinel-1 image was spliced, clipped, and resampled to ensure the same spatial resolution of the DEM and Sentinel-1 images. The second part mainly dealt with the generation of the standard training dataset for floods. First, a radar-based water index was used to segment a rough undulated waterbody boundary of the studied flood event. Then, a regional threshold method was adopted to refine the segmentation of the undulated extent of the flood in association with manual annotation and auxiliary DEM data to generate the strong label dataset by clipping VH, VV polarization, DEM, and label into tiles. The last part is preliminary for flood detection and mapping. The CNN model was firstly trained and evaluated by using previously obtained strong label datasets for flood detection and mapping; the results obtained were further processed to produce the weak label flood dataset that can be quickly expanded to the dynamic flooding database for near-real-time flood detection and mapping.

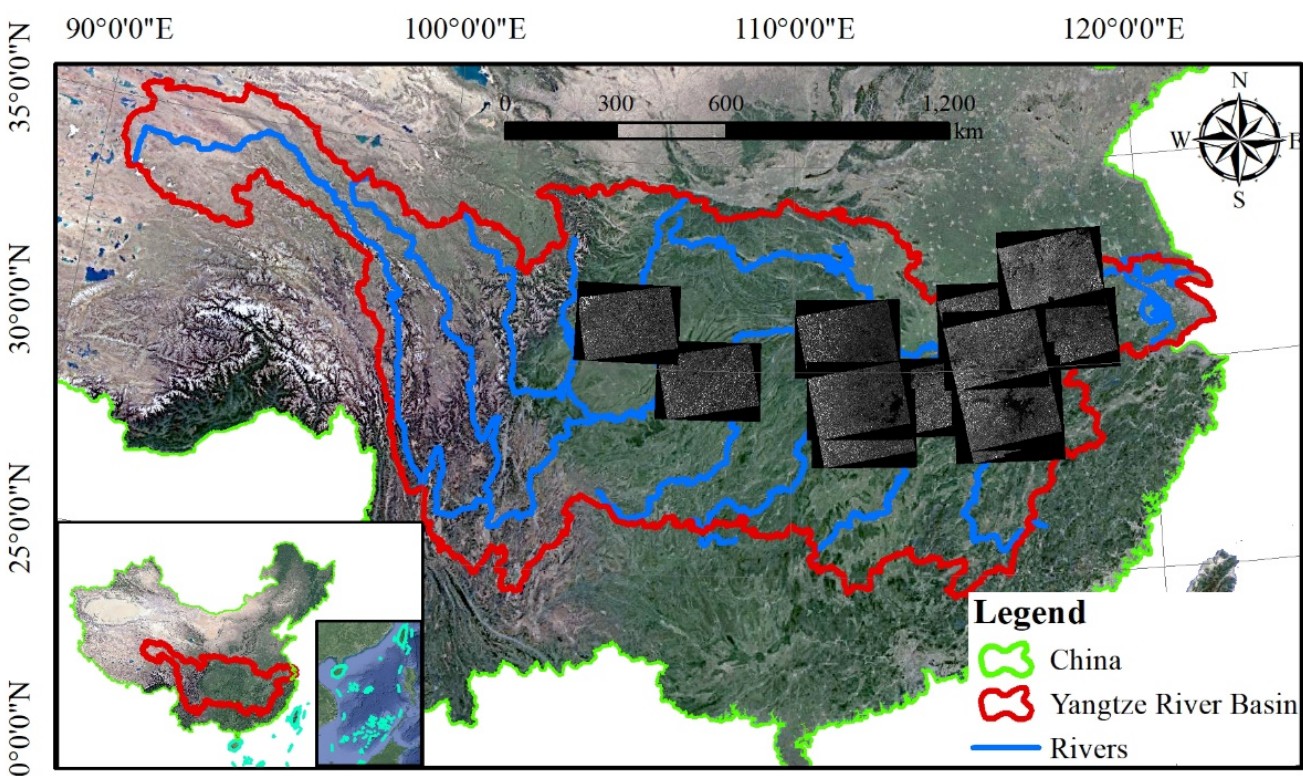

**Figure 1.** Geo-location of the YRB and the locations of the satellite data used in present study.

2.2.1. Dataset Production Method

The Sentinel-1 images acquired in each of the 8 selected flood events in the YRB were preprocessed with SNAP software. Meanwhile, DEM corresponding to the coverage of each Sentinel-1 images was spliced, clipped, and resampled to ensure the same spatial resolution of the DEM and Sentinel-1 images; additionally, a water index method used in previous studies [38] with a threshold of 0.3–0.4 for the rough segmentation of the flood waterbody was applied to extract the extent of the undulated area. Formula (1) lists the definition of the water index, where VH and VV represent polarized bands of the SAR images:

$$\mathrm{WI} = \ln(10 \times VH \times VV) - 8 \tag{1}$$

It should be noted that the results derived in this way are just the rough segmentation results with many errors. To obtain accurate deep learning labels of the flood training samples, the careful selection of as many as possible regions of interest (ROI) covering various ground objects to generate classified datasets into training and test samples were necessary. As shown in Figure 3, among 8 flood events selected, 7 flood events were used for training and testing, and 1 flood event was only used for testing. Two test datasets were obtained to test the robustness and generalization of the model. As can be seen from Figure 3, the training and test datasets include various land cover types to ensure the balance of positive and negative samples for deep learning.

Once the selection of the ROIs and the generation of the training and testing datasets were completed, the region threshold method was adopted to refine the segmentations previously derived. In the present study, we found that for hilly areas with many terrain shadows, a single segmentation threshold of 0.4 could correct most of the misclassifications in rough segmentations, while for areas such as farmland and aquaculture, a threshold of 0.15–0.2 was more appropriate. For mountainous areas with steep terrain, the mask with a slope of 10 degrees was used to further correct the effects of terrain shadows. Thus far, almost all processes were implemented programmatically in batches. The remaining small part that was difficult to solve by the threshold method was completed by manual

annotation. Figure 4 exhibits the results of fine segmentation. It can be seen from the figure that most of the mis-segmented areas by the global threshold method are well corrected.

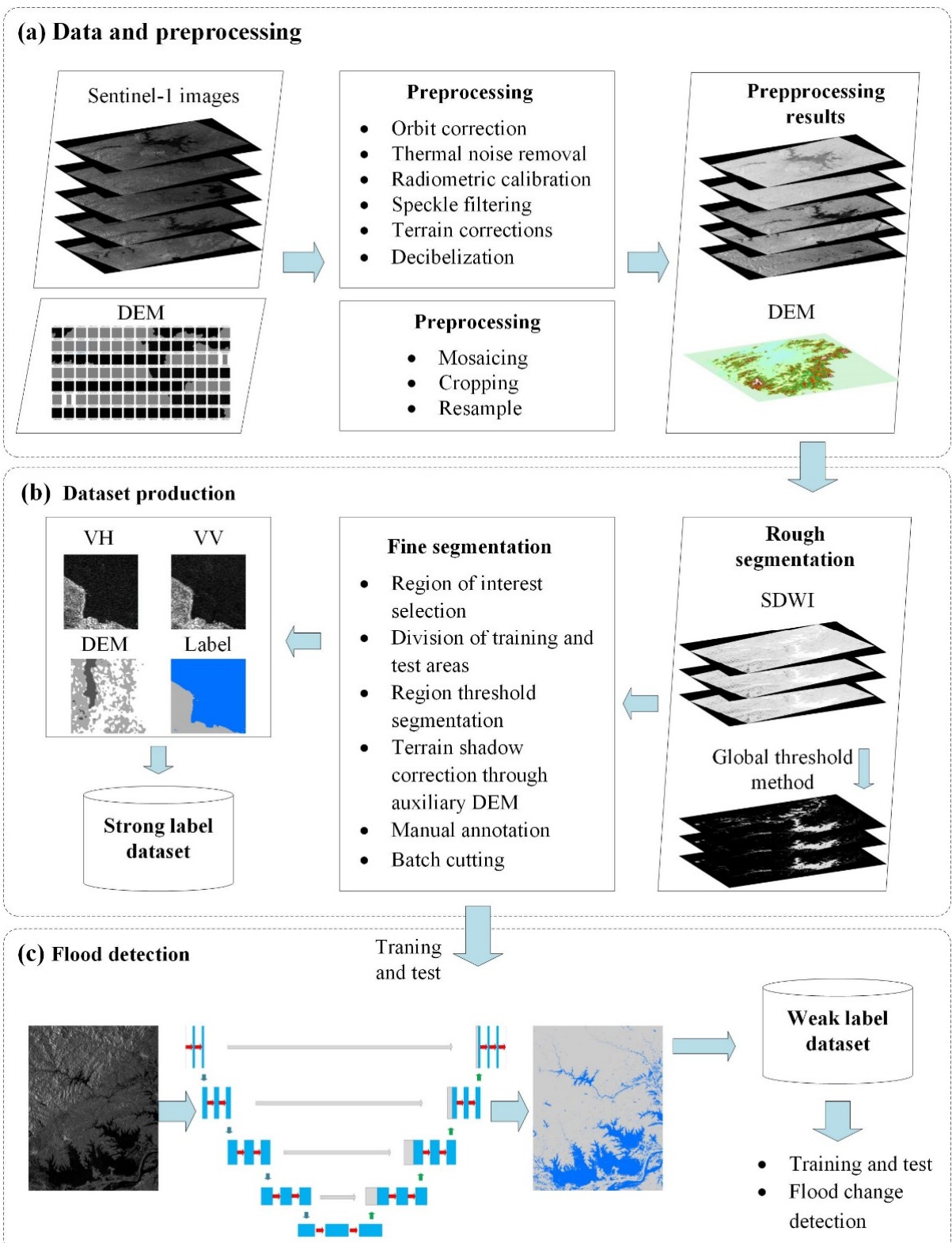

**Figure 2.** The flowchart of present study for near-real-time flood detection and mapping.

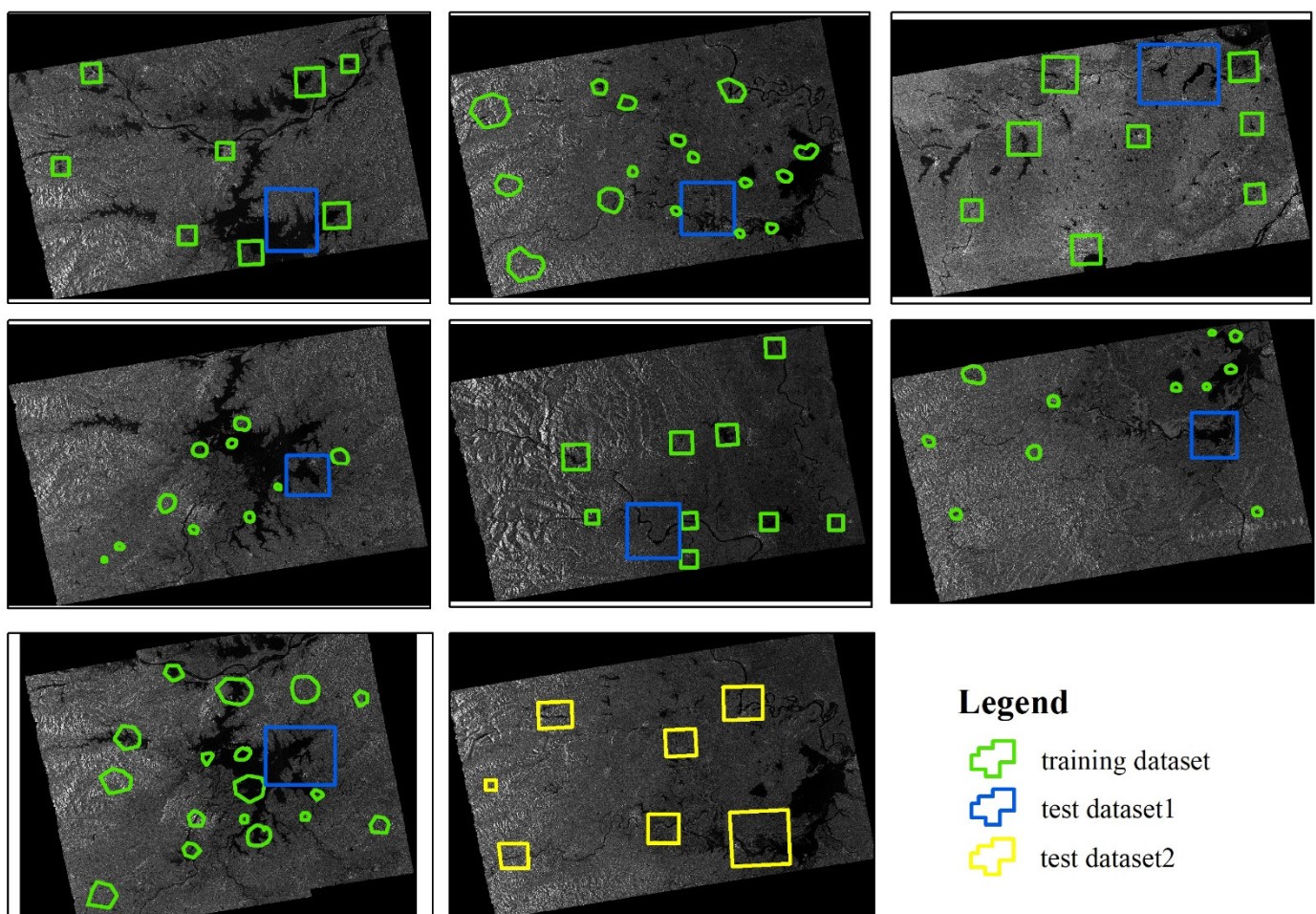

**Figure 3.** Training and test datasets selected in present study.

In the present study, 32 Sentinel-1 SAR images corresponding to the selected 16 flood events that took place in the last 2–6 years in YRB were utilized for the development of a near-real-time flood detection method. Among these, 7 flood events were used for training and testing the proposed deep learning model, and 1 flood event was only used for testing the model. In this way, one strong label training dataset and two testing datasets were obtained. Training the deep learning model requires the training data to be cropped into 256 × 256 tiles. The two testing datasets contain 13 and 14 images with 3000–5000 pixels, and were generated for near-real-time flood detection and mapping, respectively. The window cutting strategy was used for testing and application, so there was no need for image cropping. The remaining 8 flood events were used for testing the proposed near-real-time flood detection method. The tested results of the floods can be generated into weak label datasets to improve productivity of the deep learning datasets. Some examples of strong label datasets are shown in Figure 5.

### 2.2.2. Deep Learning Models Adopted for Experimental Studies

In this study, four popular deep learning models, FCN-8 [39], SegNet [40], UNet [41], and DeepResUNet [42], were adopted in order to evaluate their performances in flood detection. Fully Convolutional Networks (FCN) is the first deep learning model used for semantic segmentation. In FCN, deconvolution is used to replace the full connection layer. According to different deconvolution scales, FCN can be divided into FCN-8, FCN-16 and FCN-32. In this study, FCN-8, which has the most detailed features, was used for flood detection. UNet is the most classic and widely used segmentation network. Since many subsequent deep learning networks are proposed based on UNet, UNet structure

was introduced specifically. UNet is a typical encoding (down-sampling) and decoding (up-sampling) model. As shown in Figure 6, the encoder and decoder have a symmetrical structure, including 4 up-sampling and 4 down-sampling layers, respectively. Each sampling layer is composed of 2–3 stacked convolutional layers, and the number of convolutional layer channels is 64, 128, 256, 512, and 1024. The feature maps of the up-sampling and down-sampling layers are connected by a concatenation function to recover the details lost during the max pooling. Similar to UNet in structure, SegNet has no concatenation operation, but retains the index of max pooling in the down-sampling layer, so that the detailed features can be reconstructed more accurately. DeepResUNet takes UNet as the basic framework, but adds the residual structure of ResNet [43] and reduces the number of convolutional channels to 128, making the model more efficient.

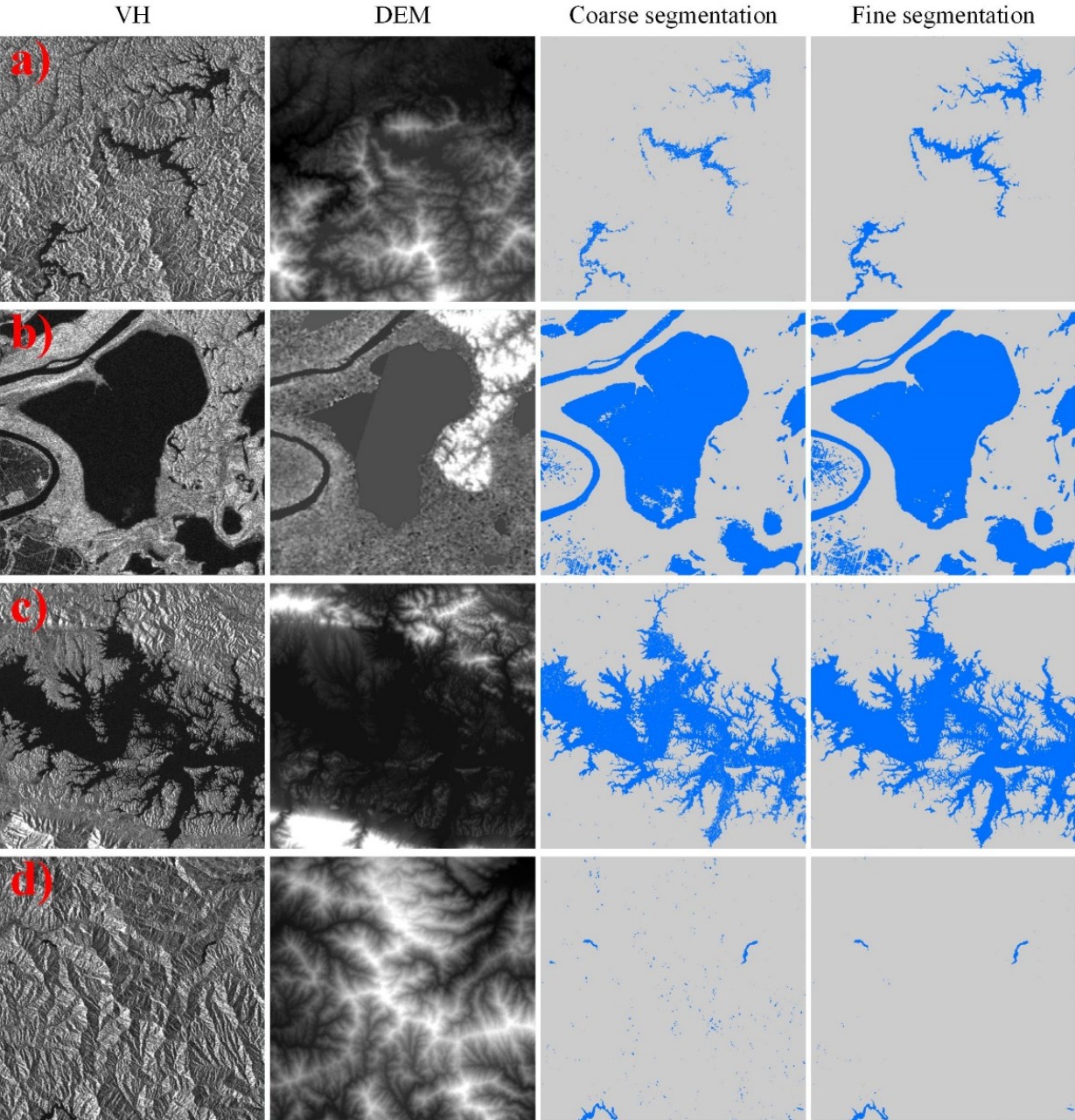

**Figure 4.** Examples of the refined segmentations for several regions. (**a**) The water bodies in mountainous areas. (**b**) The rivers and the lakes. (**c**) The lakes. (**d**) The mountain areas.

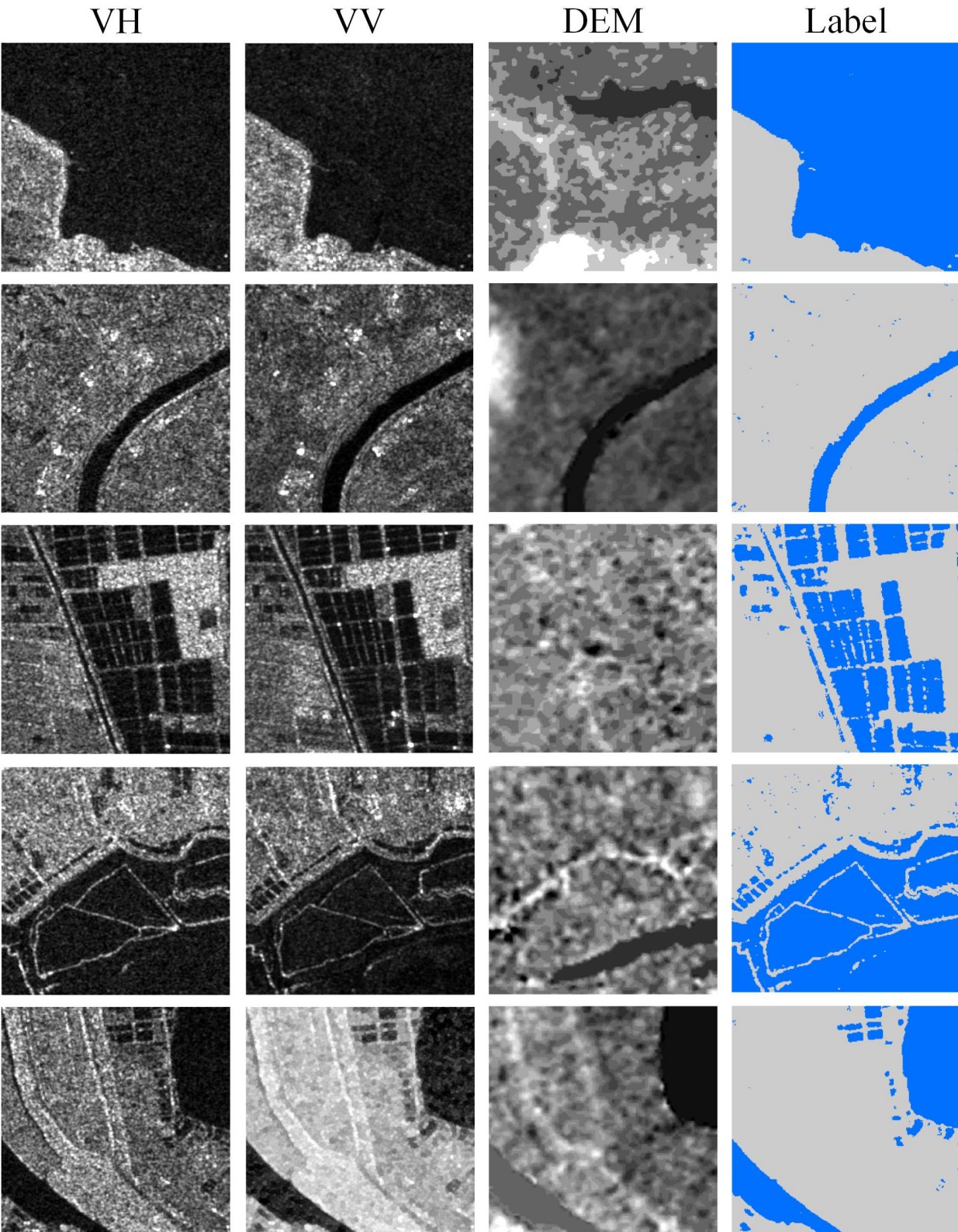

**Figure 5.** Examples of strong label dataset derived from present study.

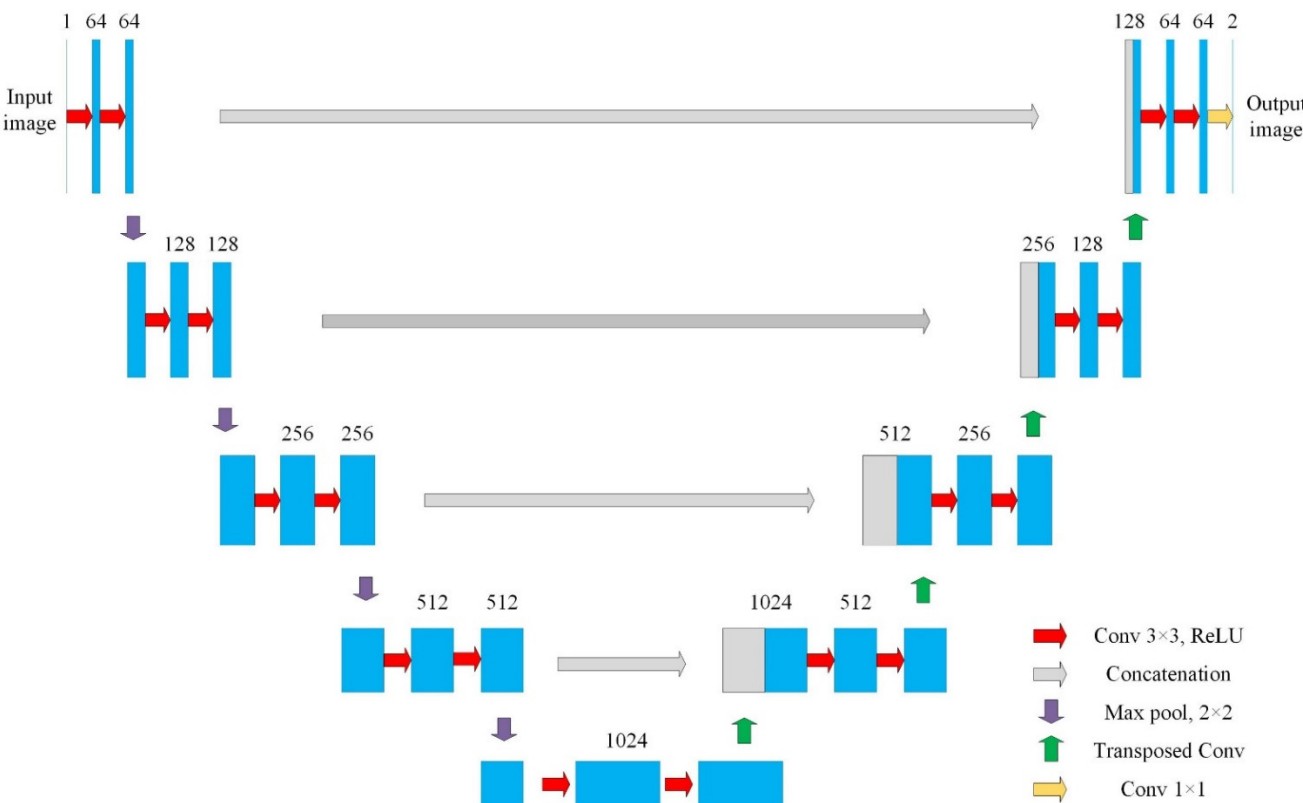

**Figure 6.** The structure of UNet.

### 2.2.3. Evaluation Metrics and Experimental Parameters

In this study, evaluation metrics, overall accuracy (OA), precision, recall and F1-score were used to evaluate the flood detection results. OA refers to the proportion of correct predictions in the total number of predictions. However, the assessment category is not balanced, and OA may be misleading. Therefore, precision, recall rate, and F1-score were also used for more objective model evaluation. For binary classification problems, the confusion matrix intuitively shows the classification of each category by the classifier. The formulas of confusion matrix and 4 evaluation indexes were given in Table 2.

**Table 2.** Metrics used in performance evaluation of the deep learning models adopted in flood detection.

| Confusion Matrix | | | |
|---|---|---|---|
| | Prediction | | |
| | | Water | No-Water |
| Label | | | |
| Water | | True Positive (TP) | False Negative (FN) |
| No-Water | | False Positive (FP) | True Negative (TN) |
| Evaluation Metrics | | | |
| Overall Accuracy (OA) | | $OA = \frac{TP+TN}{TP+TN+FP+FN}$ | |
| Precision (P) | | $P = \frac{TP}{TP+FP}$ | |
| Recall (R) | | $R = \frac{TP}{TP+FN}$ | |
| F1-score (F) | | $F = \frac{2 \times P \times R}{P+R}$ | |

SNAP 8.0 software was used to preprocess Sentinel 1 images. Arcpy and Python were used for the segmentation of coarse and fine and dataset generation. The experiments were implemented under the TensorFlow framework on an NVIDIA GeForce RTX 2080Ti GPU.

The Adam optimizer was used in computation, the training batch was set to 10, and the number of iterations was 60,000. At the same time, an exponential decay strategy was used, with 0.8 decays per 10,000 iterations and an initial learning rate of 0.0001. Table 2 lists the details of the confusion matrix used in the present study for the performance evaluation of deep learning models adopted for flood detection.

## 3. Experimental Results

### 3.1. Model Comparison Experiment

The performances of the five classic deep learning models, the global threshold method, FCN-8, SegNet, UNet, and DeepResUNet, were compared with the two sets of test datasets generated previously. As can be observed in Table 3, the global threshold method performed the worst among all the models compared, especially for recall rates, which were about 15% lower than the other models. FCN-8 had the lowest F1-score of all CNN models, and its precision was about 0.08 lower than those of other models. SegNet performed well in precision, but its comprehensive index F1-score was slightly lower than those of UNet and DeepResUNet. The performances of UNet and DeepResUNet were very close, and their accuracies were better than those of other models. The performances of all the models tested with test dataset 1 were better than those tested with test dataset 2, which can probably be attributed to the fact that test dataset 2 came from different flood events (refer to Figure 3 in Section 2.2.1).

**Table 3.** Comparisons of model performances with two different test datasets. The first line presents the models' performances with test dataset 1, and the second with test dataset 2. The values in bold indicate the highest numbers for corresponding metrics.

| Model | OA | Precision | Recall | F1_Score |
|---|---|---|---|---|
| Global Threshold Method | 0.958 | 0.977 | 0.795 | 0.877 |
| | 0.953 | 0.969 | 0.774 | 0.860 |
| FCN-8 | 0.974 | 0.943 | 0.970 | 0.956 |
| | 0.961 | 0.881 | 0.939 | 0.909 |
| SegNet | 0.983 | **0.991** | 0.953 | 0.971 |
| | 0.975 | **0.981** | 0.897 | 0.937 |
| UNet | **0.986** | 0.980 | **0.973** | **0.976** |
| | 0.978 | 0.951 | **0.942** | 0.947 |
| DeepResUNet | 0.986 | 0.985 | 0.967 | 0.976 |
| | **0.979** | 0.970 | 0.927 | **0.948** |

Flood detection results were visualized by being compared with 5812 × 4260 images derived with the test dataset 1, as exhibited in Figure 7. As can be observed, the missing detected area of flood by the global threshold method was very large, mainly distributed in the boundaries of rivers and lakes. This is attributed to the poor capability of the threshold method in dealing with the heavily noise-affected SAR images surrounding the waterbody boundaries. The wrongly detected area of flooding by FCN-8 was also large, mainly concentrated along riverbanks and surrounding lakes, as well as the small waterlogged areas. The result of SegNet was significantly better than that of FCN-8, while fewer areas were mis-detected by UNet and DeepResUNet. UNet was therefore selected for the final flood detection and mapping in present study.

### 3.2. Band Comparison Experiments

The influences of polarization mode and the application of DEM on flood detection were also experimentally investigated. The results are summarized in Table 4. It can be observed that the highest F1-score was achieved by using VH polarization alone. The

precision of VV polarization is slightly higher than that of VH polarization, but the Recall decreased considerably. However, no obvious improvements in various evaluation metrics were seen after adding DEM to facilitate the experiments. This can probably be attributed to the mountain samples in the training data set suppressing the effects of mountain shadows on flood detection, while the DEM of the middle and lower reaches of the YRB pose limited effects in nature on flood detection. When the VH, VV, and DEM channels were used as inputs for the deep learning models, the accuracies for all the models were decreased. The experiments indicated that the signal-to-noise ratio of VV and DEM bands was low, and that the VH polarization has the best effect on flood detection in the YRB.

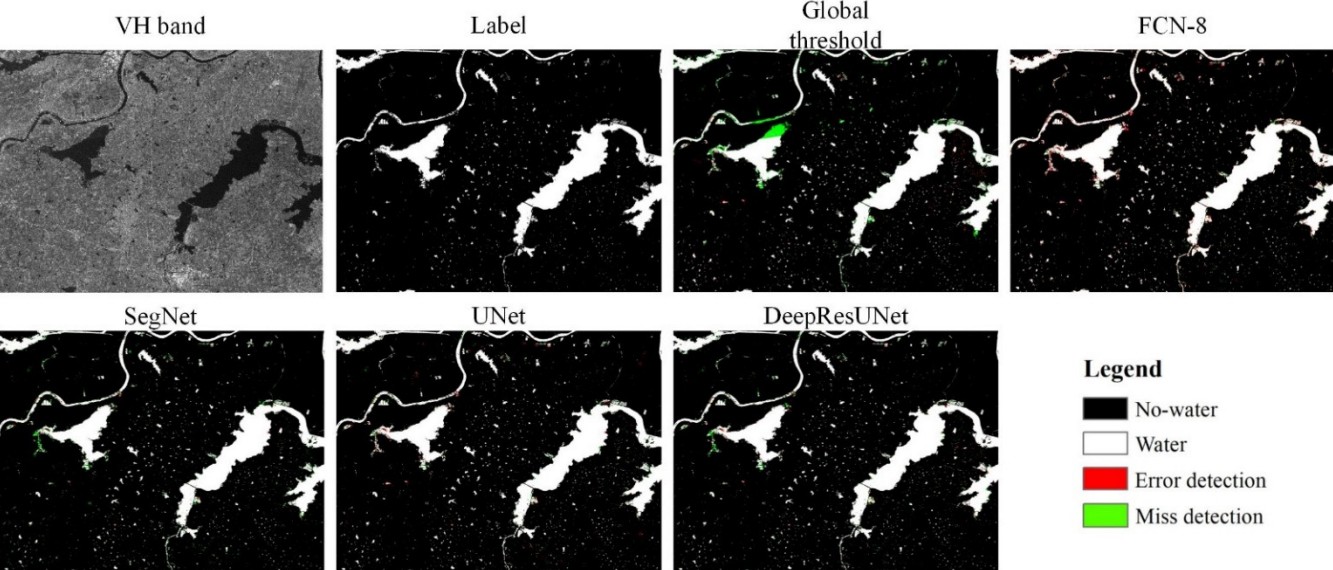

**Figure 7.** Flood mapping results of 5812 × 4260 images derived with different deep learning models with test dataset 1.

**Table 4.** Band comparison experiments on UNet with two different test datasets. The first line presents the model performances with test dataset 1, and the second with test dataset 2. The values in bold indicate the highest numbers for corresponding metrics.

| UNet/Band | OA | Precision | Recall | F1_Score |
|---|---|---|---|---|
| VH | **0.986** | 0.980 | 0.973 | **0.976** |
| | **0.978** | 0.951 | 0.942 | **0.947** |
| VV | 0.976 | **0.985** | 0.933 | 0.958 |
| | 0.961 | **0.972** | 0.835 | 0.898 |
| VH + DEM | 0.986 | 0.976 | **0.975** | 0.976 |
| | 0.978 | 0.941 | **0.952** | 0.947 |
| VV + DEM | 0.977 | 0.966 | 0.954 | 0.960 |
| | 0.964 | 0.934 | 0.886 | 0.909 |
| VH + VV | 0.983 | 0.980 | 0.961 | 0.971 |
| | 0.971 | 0.966 | 0.889 | 0.926 |
| VH + VV + DEM | 0.981 | 0.988 | 0.948 | 0.968 |
| | 0.968 | 0.979 | 0.865 | 0.918 |

Similarly, flood detection results were visualized compared with 2688 × 2248 images derived from test dataset 2 as exhibited in Figure 8 for investigating the performances of UNet with different band combinations as inputs. As can be seen from Figure 8, only a few

mis-detected or missing flooded areas existed in the map generated with UNet model with VH polarization band as input, and the errors mainly concentrated on the areas surrounded by the flooded area. The flood detection results with the VV band as input displayed poor accuracy because of many errors in river edge detection. Adding auxiliary DEM as an input did not improve the flood mapping results but introduced some noise. Band combination of VH and VV polarization as input for the UNet did not improve the performances of the model, most likely due to the reason previously analyzed.

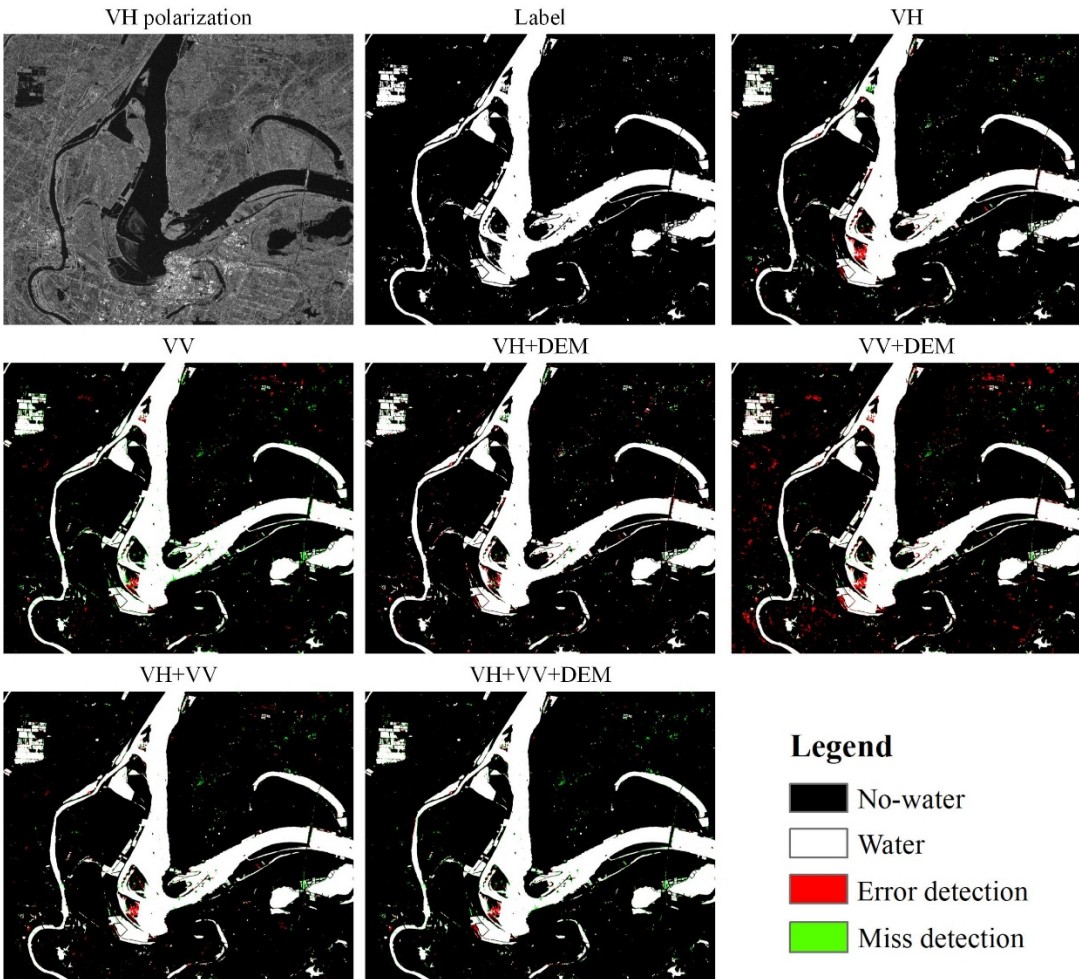

**Figure 8.** Flood mapping results of 2688 × 2248 images derived with UNet with the test dataset 2.

### 3.3. Near-Real-Time Flood Detection and Mapping

Using the UNet model trained with the strong label datasets generated previously, the near real-time flood detection and mapping were performed with the remaining eight flood events. Figure 9 presented the detection and mapping results of four flood events. The first two columns of image maps shown in Figure 9 were the VH band images acquired in each flood event and the images acquired in non-flood period, respectively. The red part indicated the flood area detected and mapped with the UNet, obtained through the difference between the detected results in flood period and non-flood period. The floods shown in Figure 9a took place in the middle reach of the YRB over the Honghu Lake near Jingzhou city, Hubei province, China. As can be seen from the figure, large areas of lakes and the main stream of the YRB have been flooded. The floods shown in Figure 9b occurred in the Chaohu Lake basin of Hefei City, Anhui Province. The area inundated by the flood was mainly cultivated land and farmland, resulting in great agricultural losses during this event. The serious floods detected with Sentinel-1 SAR images by the UNet, as presented in Figure 9c, took place in July 2017 in Yueyang city, Hunan province, while regional floods

occurred in the Dongting lake basin in Hunan province. As can be seen from the figure, the Dongting lake expanded by more than two times in area, and an extensive area of cultivated land has been inundated. The floods exhibited in Figure 9d happened in July 2020 over the Poyang Lake in Jiangxi province. During this flooding, many wetlands near the Poyang Lake were heavily affected. From the above flood examples, it is obvious that the flood areas in the YRB are mainly concentrated in the middle and lower reaches of the basin, and the floods that took place in the key areas of the Poyang Lake and Dongting Lake were especially serious. The remaining near-real-time flood detection results are shown in Figures A1–A4.

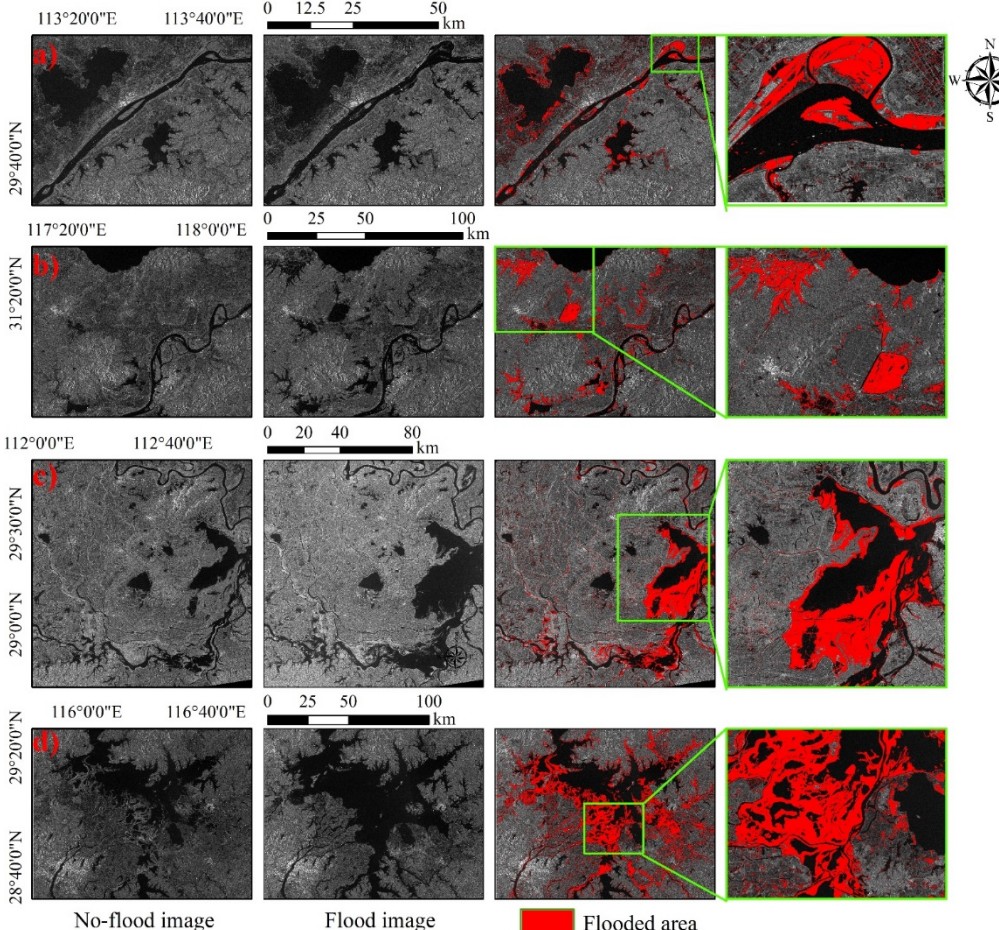

**Figure 9.** Near-real-time flood detection results obtained with Sentinel-1 SAR images by using deep learning model of UNet. (**a**) Flood detected in Honghu Lake in 2020. (**b**) Flood detected in Chaohu Lake basin in 2020. (**c**) Flood detected in Dongting Lake in 2017. (**d**) Flood detected in Poyang Lake in 2020.

## 4. Discussion

### 4.1. Weak Label Datasets Experiments

From present study, it can be concluded that the CNN-based flood detection deep learning model is efficient and fast, which is of great significance for improving the efficiency of near real-time flood detection. In practice, however, quick and efficient flood mapping technology is essential for disaster prevention and mitigation. To improve the efficiency of flood mapping, the test results in Section 3.3 were made into a weak label dataset. First of all, the detected flood image was cut into tiles with a size of 256 × 256 pixels, among which some tiles were almost fully flood-covered while some were non-flooded. The algorithm used for this processing eliminated 80% of such tiles, leaving 21,826 tiles to form the weak label dataset that consisted of partly flood-covered and partly non-flood-covered

tiles. With this weak label dataset, performances of deep learning models were evaluated and comparison experiments with different band combinations as inputs were carried out, and the results were shown in Table 5. Since the performances of those models have been evaluated previously, here we combined two test datasets to reduce the number of tables. Similar concluding remarks as obtained in performance evaluations of deep learning models can be summarized:

(1) Performances of the UNet and DeepResUNet were fairly close with each other, while FCN had the lowest flood detection accuracy;
(2) The VH polarization band as input for the deep learning models performed the best in flood detection, while the DEM had a very minor affect on the results of flood detection.

**Table 5.** Model performances with weak label datasets.

| Model | OA | Precision | Recall | F1_Score |
|---|---|---|---|---|
| FCN-8 | 0.948 | 0.897 | 0.909 | 0.903 |
| SegNet | 0.955 | 0.912 | **0.917** | 0.914 |
| UNet | **0.958** | **0.930** | 0.911 | **0.920** |
| DeepResUNet | 0.958 | 0.927 | 0.912 | 0.919 |
| **UNet/Band** | **OA** | **Precision** | **Recall** | **F1_score** |
| VH | **0.958** | 0.930 | **0.911** | **0.920** |
| VV | 0.952 | 0.914 | 0.904 | 0.910 |
| VH + DEM | 0.958 | **0.933** | 0.905 | 0.919 |
| VV + DEM | 0.953 | 0.918 | 0.902 | 0.910 |
| VH + VV | 0.957 | 0.928 | 0.910 | 0.919 |
| VH + VV + DEM | 0.955 | 0.922 | 0.908 | 0.915 |

In general, the effect of the weak label dataset on the performance of the CNN model was not that significant compared that of the strong label dataset. This could be attributed to, on the one hand, the weak label dataset having not been selected and marked manually, so the overall accuracy was lower than that of the strong label dataset; on the other hand, the weak label dataset comes from eight flood events detected with UNet model, which was completely different from the generating methods of the other eight flood events used for training and testing. Therefore, we can refer to the flood detection results with the strong label dataset to optimize the the flood detection results with the weak label dataset for quickly expanding the training dataset of flood samples.

*4.2. Change Detection Method*

The methodological logic we followed for flood detection was based on the difference between the detected waterbody extent in flood period and the natural waterbody extent in the non-flood period to determine the flooded area. In fact, this is not a complete end-to-end flood detection method. In CNN, images acquired in the flood and non-flood period can be set as input, and the label is the changed part of the waterbody extent between these two periods, so the output is the flooded area. Currently, two kinds of change detection models based on the convolutional neural network are popularized. One is to use ordinary neural network models to directly learn the features of the changes [44]. The other is to use a Siamese neural network model that uses two networks to extract features, which shares weights between the two networks [45,46]. The long short-term memory (LSTM) convolutional neural network [47] with a time series of remote sensing images as input has been proposed for change detection, which sheds light on the possibility to expand the detected changes into a standard change detection dataset in the near future. However, it is worth noting that the following issues remain as challenges:

(1)   The selection of areas with high classification accuracy to prevent noise interference;
(2)   Just like the weak label dataset, some of the unchanged data labels need to be eliminated;
(3)   The proportion of positive and negative training samples should be balanced, or a special loss function, such as dice loss, needs to be considered.

### 4.3. Novelty, Potential, and Limitations

Deep learning is a data-driven algorithm. From the perspective of training and testing datasets, the present study aims to improve the efficiency of flood detection and mapping by using convolutional neural networks for large-scale near-real-time flood detection and mapping. Four classic CNN models were used for large-scale near-real-time flood detection and mapping in the present study. In the future, by means of spatial pyramid structures, feature reuse structures or attention mechanisms, the number of convolutional kernel channels can be reduced to reduce the redundancy of the model for improving the accuracy of the model. Meanwhile, the integration of the strong label dataset and weak label dataset can effectively facilitate long time series flood disaster monitoring. Sentinel 1 satellite is constrained by its revisit cycle for effective capture of flood events. More SAR satellites are needed to enhance the generalization and applicability of datasets for large-scale near-real-time flood detection and mapping.

In some studies, optical and SAR image data are used for flood detection to improve the accuracy of the results [48–50]. The spectral information of optical data is easier to identify water bodies from than SAR data. In the future, we can find optical images with no or little cloud coverage during flooding events to expand the available datasets. The floods in the YRB are concentrated in the middle and lower reaches of the basin where the terrain is flat and DEM has limited effects. However, for flash floods, DEM are important for identifying mountainous shadows that may pose serious effects on flood detection and mapping.

### 5. Conclusions

The suddennature of flooding makes it difficult to seize the dynamic process and the extent of flooding in real time which is essential in disaster prevention and mitigation. Although deep learning is a promising technology to assist remote sensing for near-real-time flood detection and mapping, constrained by the shortage of qualified flood training and testing samples and the low efficiency of the data processing procedures involved, the performances of the available deep learning models for large-scale flood detection and mapping are far beyond the expectations. For breaking through the current predicament, a semi-automatic flood dataset-generating method was proposed to counter the problems in efficient generation of strong label datasets at first and then realize the near-real-time flood detection and mapping in the YRB by using the generated strong label flood datasets with association of CNN model. Several experiments were conducted to investigate the performances of the proposed method under various conditions. It was concluded that the VH polarization data of SAR images alone performed the best for flood detection, while the involvement of the DEM as input for the CNN posed limited effects in flood detection over the YRB. Meanwhile, the weak label dataset was generated according to the near real-time flood detection results, and experiments on near real-time flood detection with the expanded flood datasets proved that the weak label dataset positively affected the flood detection. If the procedure for weak label dataset generation can be improved (refer to the production method of the strong label dataset), the efficiency and precision of flood datasets and flood detection and mapping results can be greatly improved. In short, from the perspective of datasets, this study proves that CNN has great potential for high-efficiency flood dataset generation as well as in near-real-time flood detection.

**Author Contributions:** X.W. and Z.Z. designed this study. Z.L., B.A. and R.L. completed data collection and preprocessing. X.W. wrote this manuscript. Z.Z., W.Z., S.X. and J.T. revised the manuscript. All authors have read and agreed to the published version of the manuscript.

**Funding:** This study was jointly financed by Major science and technology project of Ministry of Water Resources [Grant No. SKS-2022008] and the Key R & D and Transformation Program of Qinghai Province [Grant No. 2020-SF-C37].

**Data Availability Statement:** The Sentinel 1 image used in the article can be downloaded from ESA through product ID (Link: https://scihub.copernicus.eu/dhus/#/home accessed on 5 April 2023).

**Conflicts of Interest:** The authors declare no conflict of interest.

## Appendix A  Near-Real-Time Flood Detection Results

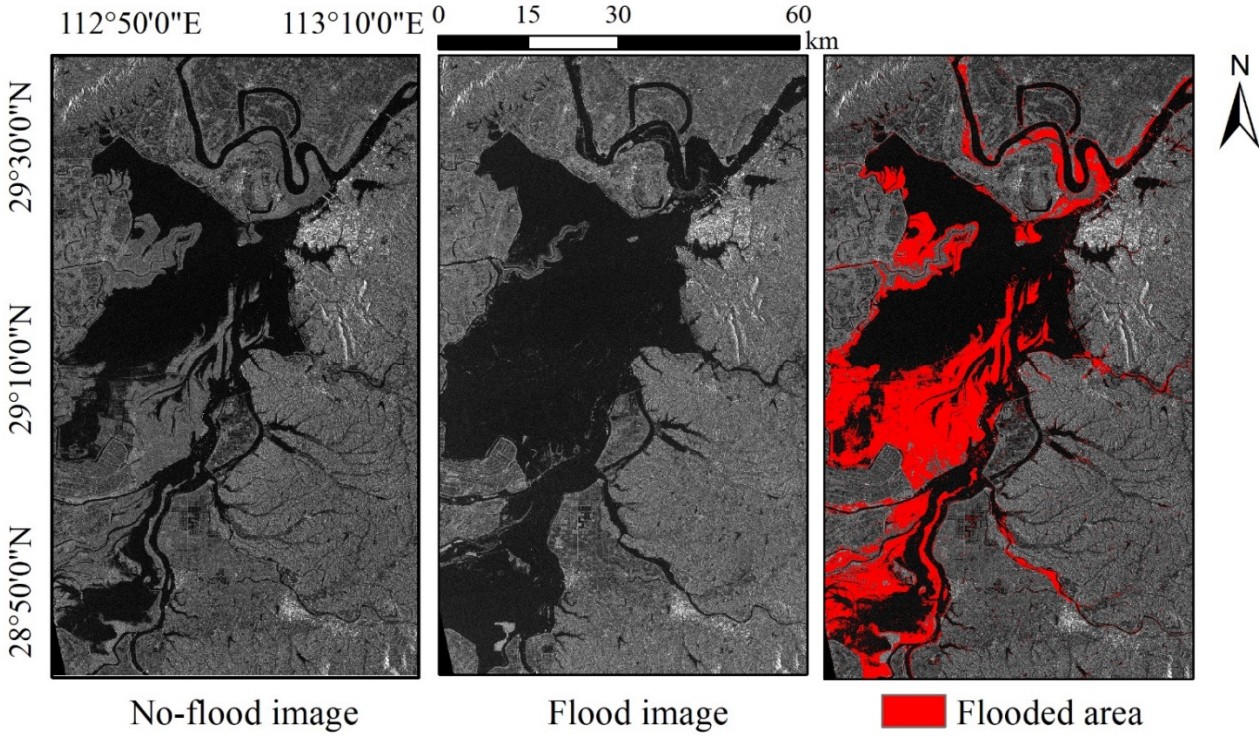

**Figure A1.** Flood detected in Dongting Lake in 2020.

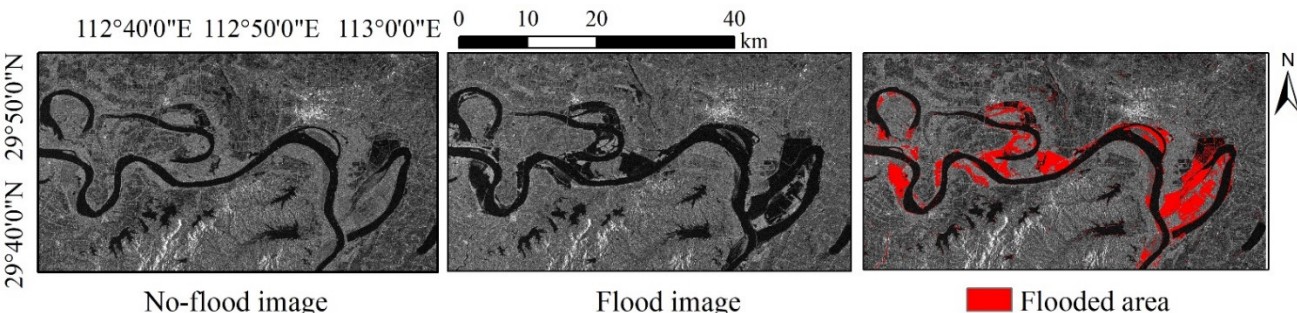

**Figure A2.** Flood detected in the middle reaches of the Yangtze River in 2020.

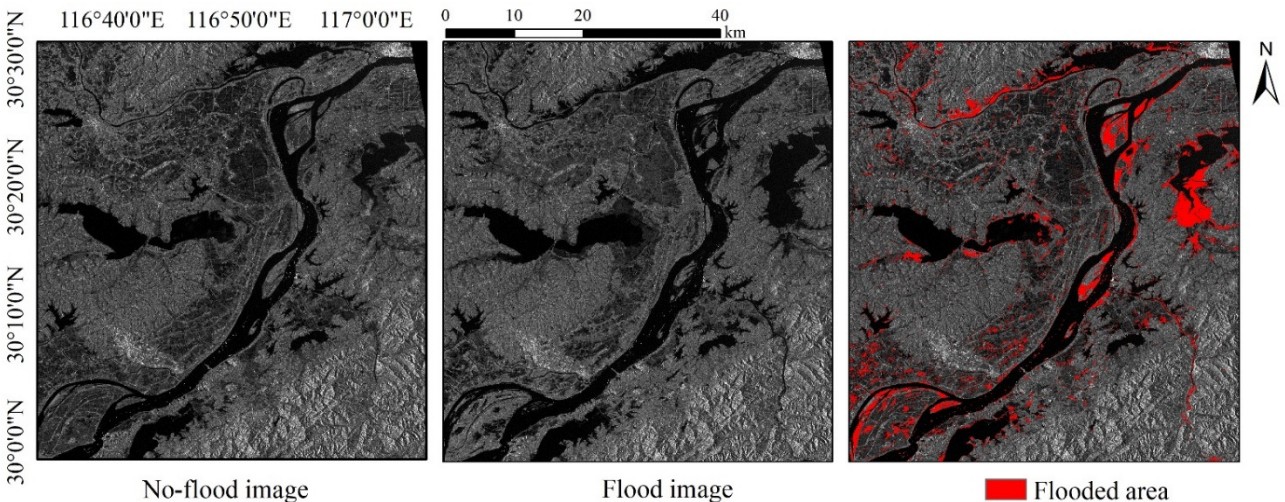

**Figure A3.** Flood detected in Shengjin Lake in 2020.

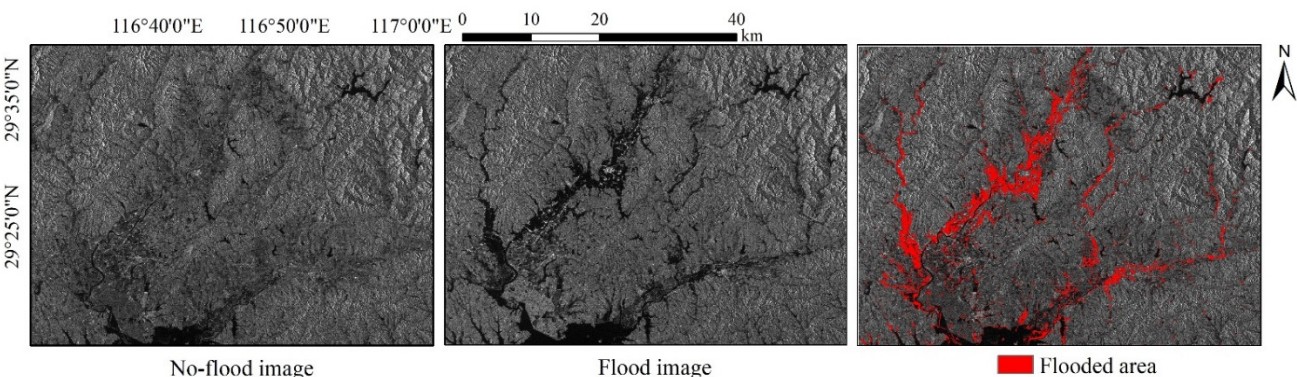

**Figure A4.** Flood detected in Xinmiao Lake in 2020.

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
