# Peer review of "A Near-Real-Time Flood Detection Method Based on Deep Learning and SAR Images"

_remotesensing, doi:10.3390/rs15082046_

Round 1

Reviewer 1 Report

The paper by Wu et al. is tackling a growing topic in using SAR for detecting flood. The authors did a good job in developing the methodology and presenting their results. The main issues of the paper are 1) English, and 2) not sharing the codes. Here are some more detailed comments.

The English needs to be improved especially with regards to the tense of the verbs.

Please spell out all acronyms in the abstract. Also please remove some of the introductory language from the abstract and add more on your results.

Introduction, the second line (line 45): it seems the “46” needs to be removed.

Same comments on “49” in line 47.

Line 50, I think the authors meant fatalities not facilities.

What is RMB?

The Authors needs to discuss the advantages of using remote sensing in general over models and ground truthing sensors in flood mapping and prediction.

Please provide more citations for this sentence “However, the threshold method can’t deal with complex nonlinear problems, and is lack of spatial consistency and vulnerable to noise interference.”

In a formal writing “can’t” should be avoided and “cannot” should be used.

The three challenges mentioned for the near-real-time flood detection are all almost the same which is lack of big data. The authors should explain how having a multi-event dataset could help the accuracy and reliability of the outcome.

Please provide a higher quality map for all figures.

Methods: Please provide a link for any code developed to preprocess and analyze the data in this study.

Please mention what software and what version was used for this study.

The authors mentioned: “a water index method referred to previous studies [37] with a threshold of 0.3-0.4 for rough segmentation of flood waterbody was applied to extract the extent of the undulated area.” However, a big part of this study is about getting rid of the thresholds. How could you make the choice of such threshold as a part of your automated system? Maybe a calibration factor?

Lines 201-214 about all datasets should be mentioned earlier.

Please spell out acronyms for the first time in the entire paper.

Author Response

Thanks very much for your excellent and professional comments on our manuscript. We appreciate your valuable comments and suggestions to improve it. We have revised the manuscript according to your comments, please refer to the attachment

Reviewer 2 Report

Comments to the author

In this study, the authors studied the detecting method of the flood with deep learning and remote sensing data for the Yangtze River Basin. They used the several classic CNN models (FCN8, SegNet, UNet et al.) to check the flooding. In general, the manuscript is interesting for the readers. The procedures and results are well described and logically presented. This method will be helpful to detect the flooding of the YRZ.

However, there are somethings to improve the manuscript as follows.

1. Need the detailed explanation for the input data, which are the remote sensing data (SAR).

2. Show the comparison of the several CNNs in the theoretical background.

3. Explain the reasons, why the researchers choose the confusion matrix to show the performances of DL methods.

4. In the discussion and conclusion, it is a little bit lengthy and difficulty for the readers to understand. Please rewrite those parts to improve the readability.

5. Recommend for getting the professional English proofreading service to improve the readability for the reader.

With this, I recommend that the paper be returned for a minor revision to remedy this defect.

Author Response

Thanks very much for your excellent and professional comments on our manuscript. We appreciate your valuable comments and suggestions to improve it. We have revised the manuscript according to your comments, please refer to the attachment.

Reviewer 3 Report

This manuscript does an excellent job of combing deep learning and SAR dataset for flood detection. Applying five deep learning models and comparing their results provide evidence of their capability for flood studies. Although such kind of trial is significant and meaningful for applying DL to flood studies in a technique way, how to specify the flooded region is not sufficient. The manuscript is very well prepared in structural and English writing, so minor revision is necessary.

Minor comments:

The flooded area given by this work is the difference between the “flood” and “non-flood” periods. If the “non-flood” period is a dry period for a lake or other waterbody, then the results would be overestimated. Is that possible to distinguish the “non-flood” and dry period using the current approach?

Can authors explain how to divide the 16 flood events into three categories? For example, how does the selection of the test dataset affect the results? Will the numbers in Table 4 change if the test datasets are different?

What is the physical meaning of the performances the selected DL methods give? Why some of them are good, and some are not? If the flood training events were changed, would there be differences?

For Fig.1, is that possible to show the locations and shape of the lake? Since several lakes were listed, what are the satellite images shown in Fig.1 represent? Are they related to the waterbodies mentioned in Table 1?

Please explain what VH and VV are for their first appearance in the article.

Please unify F1-score and F1_scoure.

Author Response

(The authors gave the same response as above.)
